# REVIEW ARTICLE

# Design, synthesis and applications of responsive macrocycles

Jingjing Yu[1], Dawei Qi[1] & Jianwei Li [1,2✉]

Inspired by the lock and key principle, the development of supramolecular macrocyclic chemistry has promoted the prosperous growth of host-guest chemistry. The updated induced-fit and conformation selection model spurred the emerging research on responsive macrocycles (RMs). This review introduces RMs, covering their design, synthesis and applications. It gives readers insight into the dynamic control of macrocyclic molecules and the exploration of materials with desired functions.

Host–guest chemistry grew out of the lock and key principle by Emil Fischer. The principle was formulated in 1894 by his discovery that glycolytic enzymes could selectively recognize stereoisomers of sugars. He hypothesized that only when the geometry of the enzyme (lock) was exactly complementary with that of the substrate (key) could the recognition take place and subsequently trigger the catalytic reactions. This model not only helps us understand the basics of biochemistry but also encourages chemists to explore functional host molecules acting like the 'lock'. The first family of artificial host molecules, crown ethers, was discovered by Charles Pederson in 1967[1]. They were cyclic molecules, and certain metallic atoms can be bonded in the middle of the ring of these molecules. Since then, supramolecular chemists have developed many macrocycles, including natural molecules like cyclodextrins[2] and synthesized ones, including cucurbiturils[3], pillararenes[4], asararenes[5], calixarenes[6], resorcinarenes[7] and other novel supramolecular macrocycles[8–10]. These macrocycles and their derivatives have been extensively investigated so chemists are able to achieve structure specific and highly selective recognition properties, which provide opportunities for exploring advanced applications in sensing[11], transport[12], catalysis[13] and drug/gene delivery[14].

However, the lock and key principle was updated by the induced-fit and confirmation selection model to explain the conformational change of the active site of enzymes by ligand binding. Because of this, solely investigating the well-studied static macrocycles is unsatisfactory. Colleagues have started to pay attention to responsive macrocycles (RMs). A RM is a type of macrocycle that has a confirmation and/or property that could be controlled and switched on and off by using external stimuli. RMs could give rise to dynamic cavities where various guest molecules or different states of the same guest molecule lodge, providing more possibilities to control the binding of the guest molecules by changing environments. On one hand, this will

[1] MediCity Research Laboratory, University of Turku, Tykistökatu 6, 20520 Turku, Finland. [2] Hainan Provincial Key Lab of Fine Chem, Key laboratory of Advanced Materials of Tropical Island Resources of Ministry of Education, Hainan University, Haikou 570228, China. ✉email: jianwei.li@utu.fi

give a more realistic presentation of biological functions in synthetic systems; on the other hand, macrocycles may be obtained that could perform more complex functions (i.e. cascade binding).

Unlike previous reviews focusing on either photo-RMs[15–17] or strategies for the controlled binding of organic guests[18], this paper will give readers a tutorial review about RMs. First, it will focus on the design of RMs that are operated by a single stimulus or combined stimuli, such as light, pH, and redox. Moreover, as the cyclization reaction is one of the most challenging steps to obtain the macrocycles, general strategies for the formation of the ring will be introduced. The power of RMs will also be shown by highlighting their latest application in controlling guest release, catalysis, and biomedicine. Finally, future opportunities and challenges will be discussed.

## Design

Generally, RMs are designed by introducing responsive moieties into their ring-like frameworks. In only a few examples, responsive fragments are appended to the framework to design RMs. With the two strategies, many RMs controlled by external stimuli, such as light, pH and redox, have been developed. Up to now, most RMs have been operated by a single stimulus, but the responsiveness of macrocycles could also be programmed by the introduction of several stimuli into the same macrocycle. The following section will be discussed according to the responsive type involved.

## Single-RMs

*Photo-RMs.* Light is one of the ideal candidates for stimulus because of its extraordinary nature, including its clean property, noninvasive mode, remote control ability, tunable wavelength and convenient access[19,20]. The input light energy can help a photoactive unit transform into different states, which further influences the property of the macrocycle, including its geometry, spectra absorption and solubility.

Azobenzene (Azo) is well known for its isomeric structures, which can be switched between the *trans* and the *cis* form by using alternate irradiation with UV and visible light or heating[21,22]. Such a switchable property makes distinct changes to its geometry: (i) the molecular length is shortened from 9 to 5.5 Å, and (ii) the molecular conformation turns into nonplanar from planar. Thus, Azo units have been widely used to design photo-RMs whose geometric structures could be controlled by light.

Sforazzini and co-workers synthesized a cyclic compound **1** consisting of an Azo unit, a bithiophene group, and a ten-carbon alkene chain[23]. As shown in Fig. 1a, the Azo was orthogonally connected with the bithiophene fragment due to the steric effects by the two methyl groups in the meta position on the Azo unit. The two sides of the Azo-bithiophene conjunction were further linked by the alkene chain. The alkene chain could mechanically transfer the motion of the Azo unit to that of the bithiophene backbone. When the Azo unit was a *trans* form, the dithiophene fragment were forced to twist out of coplanarity, which restricted the communication of the π-orbital of thiophenes. Under irradiation by 350 nm light, the *trans* Azo moiety was changed to its *cis* form, thereby inducing the two thiophene units almost in the same planarity and switching on the π-conjugation of the dithiophene moiety. The *trans* form could be recovered by thermal relaxation of the Azo moiety with a half-life ($\tau_{1/2}$) of ca. 4.5 days. However, such long thermal stability of the *cis* isomer was not observed for a noncyclized Azo analogue.

The introduction of more Azo units into a macrocycle may increase the structural symmetry of the macrocycle and offer more possible isomerized structures. Recently, Yuan et al. have designed a macrocycle **2** containing two Azo units[24]. The macrocycle **2** is not a three-dimensional box-like structure as usual[25,26], but more planar (Fig. 1b). The planar conformation resulted from hydrogen bonding, which restricted the rotation of the phenylene bridges connected to two Azo moieties. Because of the inward electron-rich carboxyl groups, host **2** was capable of capturing electron-deficient cationic guests G2 (bipyridinium salts). Interestingly the binding stoichiometry was an exceedingly rare 2:1 between the host and the guest. Upon exposure to UV and blue light, macrocycle **2** could be switched to three different isomers and revealed a quantitative guest release and capture function.

Clearly, more Azo units can produce more isomers of the macrocycle. However, photoisomerization performance may also highly depend on the ring strain of the macrocycle. Wegner et al. designed three Azo macrocycles consisting of two (**3**), three (**4**) and four (**5**) units of Azo (Fig. 1c)[27]. The macrocycle **3** could not be photoswitched under neither light nor heat because of the high ring strain. With one more Azo unit incorporated, macrocycle **4** could be switched into four isomers. One of the isomers presented remarkably high thermal stability. Although macrocycle **5** showed a photoisomerization behaviour similar to **4**, the identified isomers had lower thermal stabilities than those of **4**.

Diarylethenes (DAEs) have a different photochemical property than the Azo units. They can perform reversible photocyclization reactions between an open and a closed ring state under irradiation by using UV and visible light[28]. The open and closed forms have vastly different absorption spectra. Zhu and co-workers reported a DAE-based metallacycle **6** through the self-assembly of metal-ligand coordination (Fig. 1d)[29]. A photoactive antiparallel (*ap*-) pyridyl-substituted DAE ligand (*ap*-PY) can quantitatively form the *ap*-[3 + 3]-**6** metallacycle with 120° di-platinum-(II) acceptors (DP) through coordination self-assembly. Upon UV light irradiation, the *ap*-[3 + 3]-**6** metallacycle was converted to the *c*-[3 + 3]-**6** metallacycle, accompanied by a distinct colour change of solution from colourless to red. Apart from the excellent photochromic property, the metallacycle also exhibited an unprecedented photochemical conversion. The three *ap*-PY units can convert to the three closed forms *c*-PY simultaneously without undergoing a step-by-step transformation.

Spiropyran is a photo-switchable unit similar to DAE. It can also be isomerized from a closed spiro form (SP) to an open merocyanine form (MC). However, isomerization also makes changes in not only colour and structural conformation but also solubility and polarity[30]. Stang and co-workers reported a metallacycle **7** whose photo-responsiveness was achieved by spiropyran units in the backbone[31]. The metallacycle **7** was formed by the coordination between the spiropyran-functionalized dipyridyl donor and an equivalent 180° di-Pt(II) acceptor **8**. Interestingly, the **7** could not only reversibly perform shrinking and swelling of topology along with colour changes during the photoisomerization process, but it also exhibited a reversible acidochromic property during the protonation and deprotonation process of MC open form (Fig. 1e). Although the cyclic structure was mainly retained during the photoreversal process, the fatigue resistance was affected by the metastable coordination bonds.

These examples clearly demonstrate that the incorporation of photo-switches into a macrocycle could produce photo-RMs. The number of switchable units could increase the complexity of the macrocycle and have a significant impact on the ring strain that further decides the property (i.e. thermal stability) of the isomeric macrocycles. Apart from the change in stability and geometry, the spectra absorbance, polarity and solubility of the macrocycle may also be altered based on the photo-switch chosen for the design,

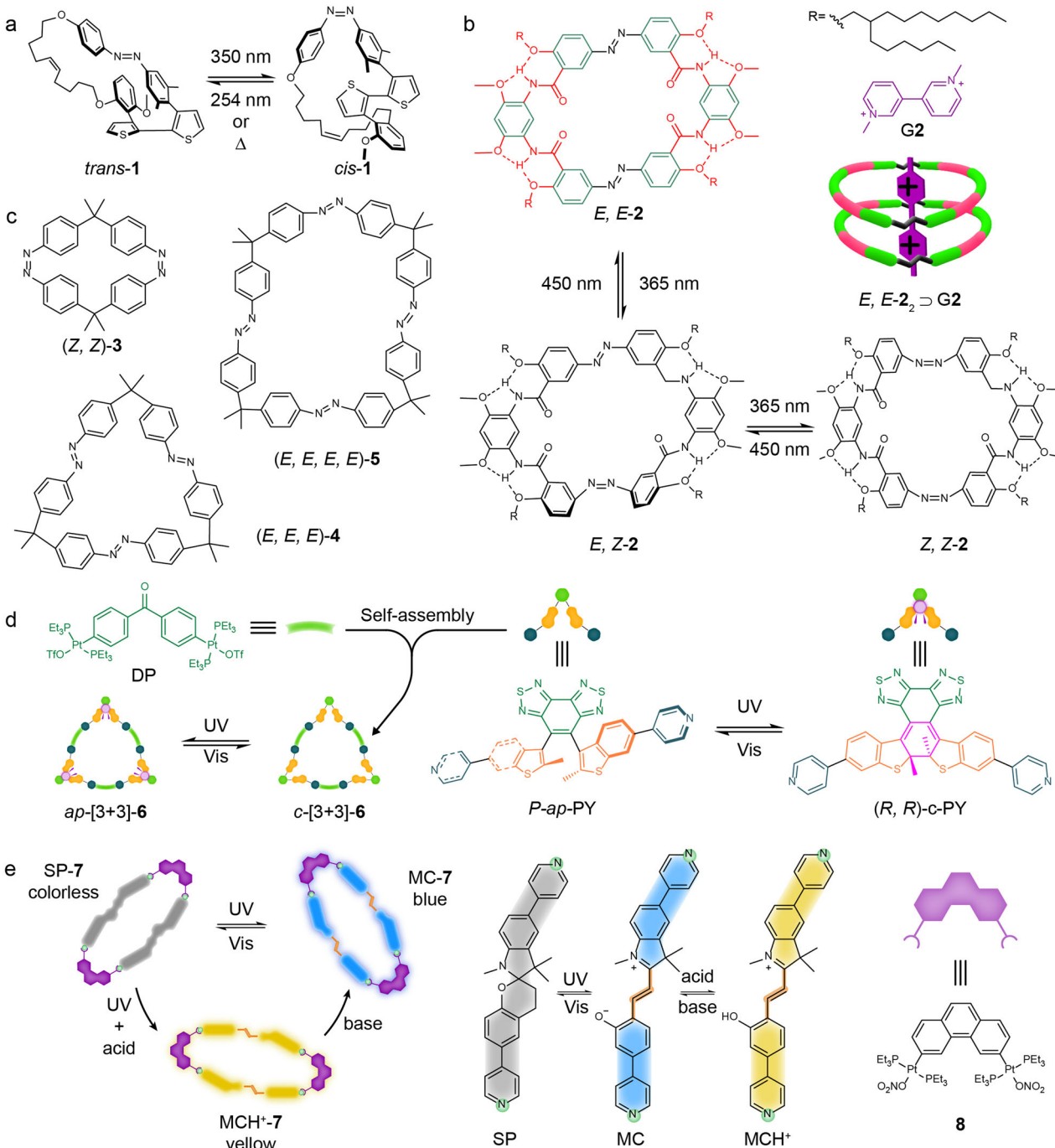

**Fig. 1 Photo-responsive macrocycles. a** planar twist of the inside connected dithiophene unit photoinduced by the azo-based macrocycle **1**[23]. **b** photo-tunable geometries of a hydrogen-bonded azo-macrocycle **2** and its specific binding model (2:1) with guest molecule G**2**[24]. **c** the chemical structures of biazobenzophane **3**, triazobenzophane **4**, tetraazobenzophane **5** contained different azo group numbers inside respectively[27]. **d** the chemical structures of donors *P*-ap-PY, *(R, R)*-c-PY and acceptor DP and photoinduced chiral switching process of DAE-based metallacycle from *ap*-[3 + 3]-**6** to o-[3 + 3]-**6**[29]. **e** the photo-controlled topology and acidochromic property of metallacycle **7**[31], the chemical structures and reversible isomerization of the spiropyran-functionalized dipyridyl donor and di-Pt(II) acceptor **8**.

providing a rich toolbox to design the photo-RMs and light-responsive materials. In addition to the photo-switches, some photoactive groups, such as anthracene[32], coumarin[33] and diacetylene[34], could perform dimerization or other photochemical reactions under light irradiation, which also enables the macrocycle response. As there are limited examples for these functional units, a detailed discussion will not be given here. For readers interested in them, please check the related references[32–34].

*pH-RMs.* The change of pH could be an effective tool to switch a reaction or a specific property of a molecule on and off[35]. Such significant changes triggered by pH have inspired supramolecular chemists to develop pH-RMs. Generally, pH-RMs are designed by the decoration of macrocycles with acid-base sensitive sites.

In 2017, Yoshizawa et al. reported an acid-base responsive polyaromatic macrocycle **9** consisting of four pH-responsive acridinium units alternatively bridged by *meta*-phenylene and *meta*-biphenyl spacers (Fig. 2a)[36]. The tetracationic framework of

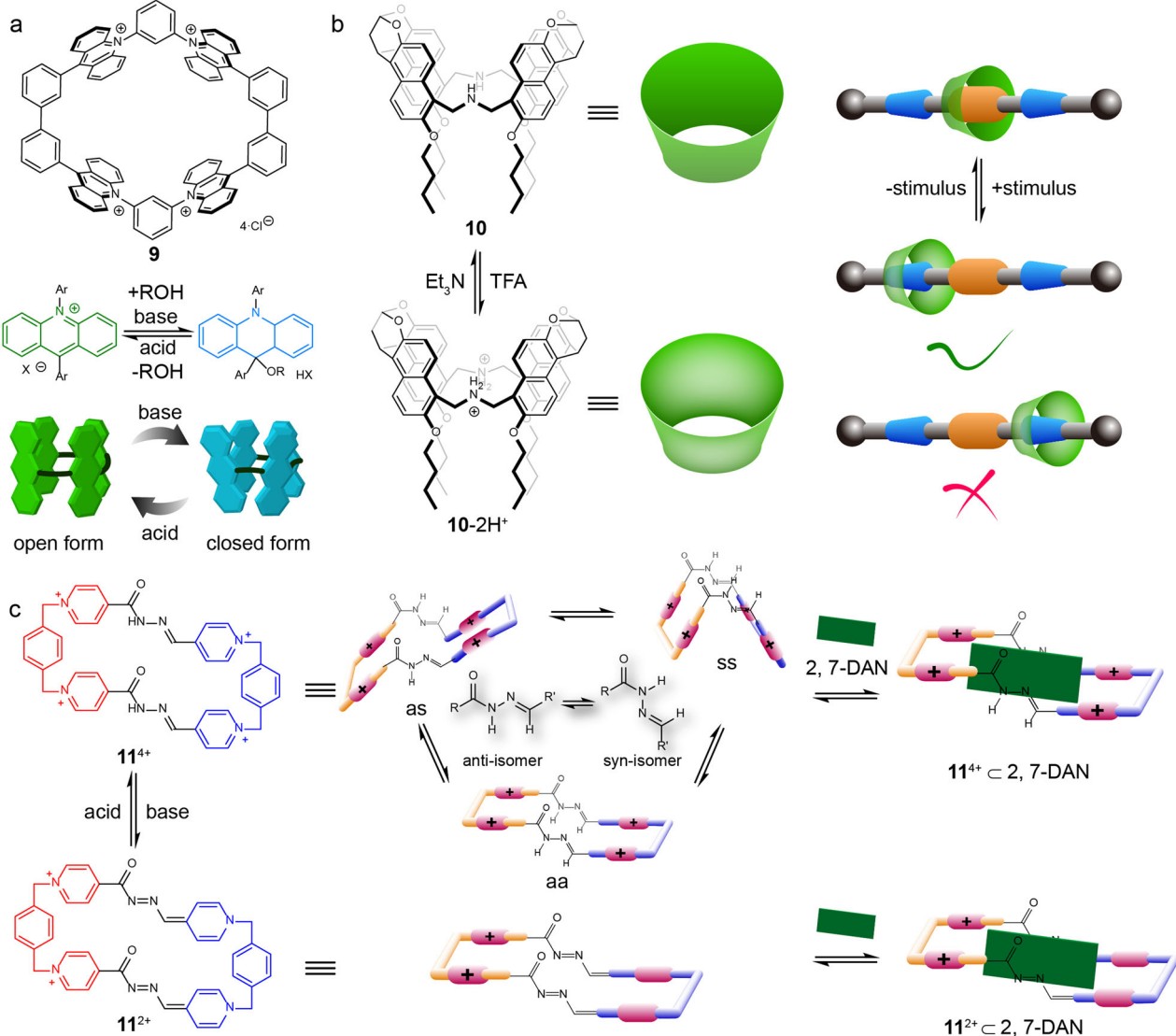

**Fig. 2 pH-responsive macrocycles. a** the chemical structure and the switchable open/closed process of polyaromatic macrocycle **9**[36]. **b** the chemical structure of the macrocycle **10** and its directional shuttling motion in a [2]rotaxane controlled by pH[37]. **c** the pH-responsive configuration change of three rotamers of the macrocycle **11**$^{4+}$ and their selective binding with the guest 2,7-DAN[38].

the macrocycle **9** ensured it was water-soluble, and the repulsion of the four positive charges made the cavity of the macrocycle **9** in an open form. The increase of pH induced the nucleophilic addition reaction to acridinium units, and the **9** became neutral. As a result, the cavity of the macrocycle was closed. The closed cavity could also be recovered by adding acid into the solution. Such a reversible change of the cavity controlled by pH enabled the **9** as a versatile host molecule that could reversibly bind small molecules, and longer molecules.

Jiang's group designed a three-dimensional pH-responsive macrocycle **10** with a cone-like inner cavity that contained four naphthalene panels connected by two secondary amine sites (Fig. 2b)[37]. The macrocycle **10** could be reversibly switched between the electron-rich state and the positively charged state **10**-2H$^+$ through protonation and deprotonation of the secondary amine groups. Macrocycle **10** can could undergo a shuttle-like motion along a rod-like compound driven by the electrostatic repulsion controlled by the protonation and deprotonation of the cyclic part. These results reveal that the pH-RMs could be significant elements for the design of smart molecular machines.

Recently, Peinador and García reported a macrocycle **11**$^{4+}$ that was synthesized by linking a pair of complementary hydrophilic bis(pyridinium)xylylene tweezers by using two acyl hydrazone bonds (Fig. 2c)[38]. Owing to the rotational isomerization of acyl hydrazones, the macrocycle **11**$^{4+}$ presented different states at various pH values in water. It had three different rotamers (aa, as, ss) at acidic condition, while the basic condition turned the macrocycle **11**$^{4+}$ into less positively charged **11**$^{2+}$. The macrocycle **11**$^{2+}$ could recognize a π-electron-rich guest (2,7-dihydroxynaphthalene, 2,7-DAN) through π–π and C–H···π interactions. When an excess of 2,7-DAN was added to the acidic solution of the macrocycle **11**$^{4+}$, only the aa rotamer remained. Thus, the conformation of the macrocycle **11**$^{4+}$ was responsive to pH changes, and the conformation could be locked by a guest molecule at the corresponding pH value.

Thus, the change of pH could protonate or deprotonate a macrocycle, which changes its charge and chemical properties, such as optical properties, hydrophobicity or the reactivity of a substance. Consequently, we can alter the shape, movement and the binding ability of the macrocycle.

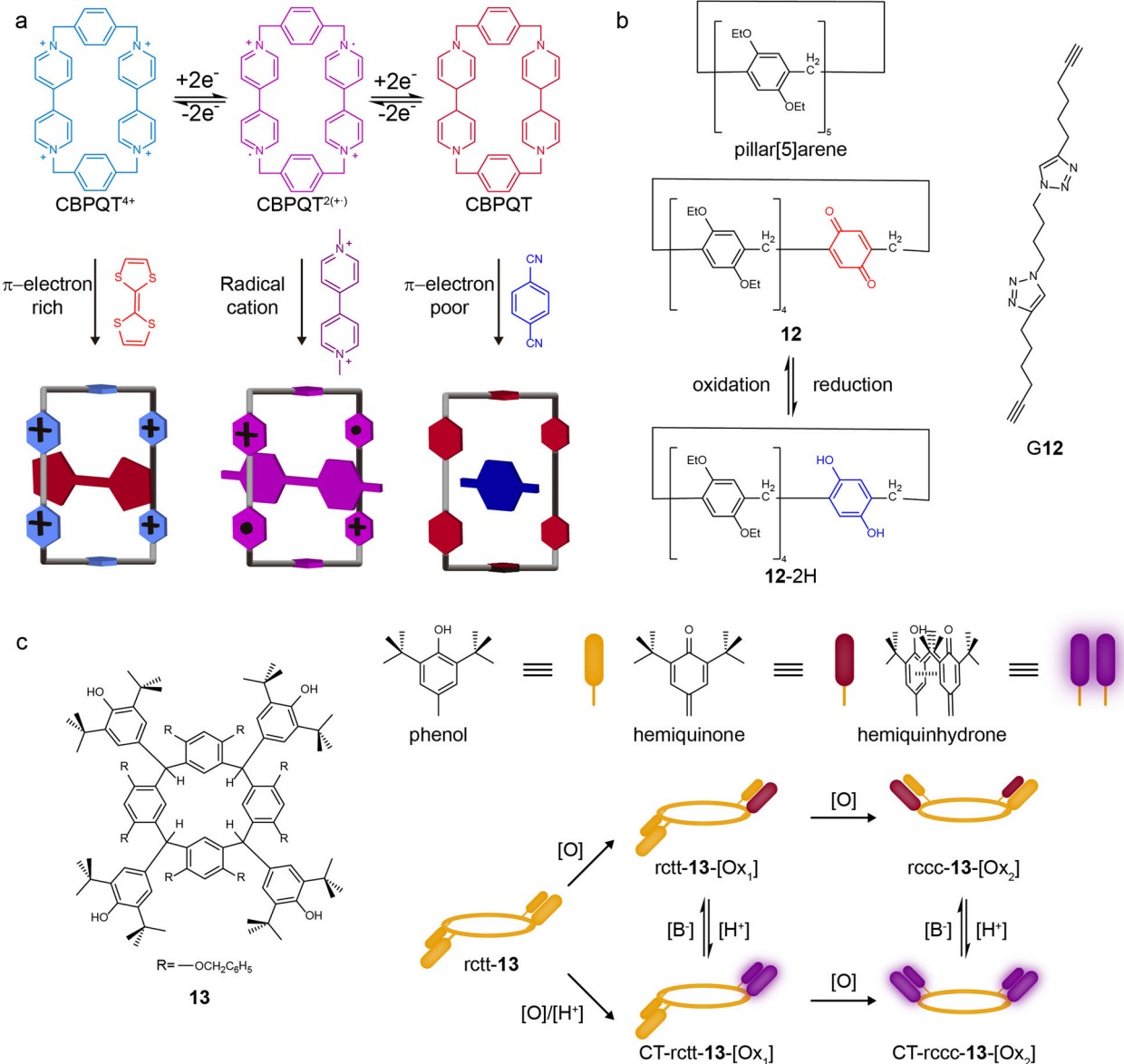

**Fig. 3 Redox-responsive macrocycles. a** the chemical structures and the specific guest-binding behaves of CBPQT[4+] macrocycle at different redox states[41]. **b** the structural changes of benzoquinone-contained pillar[5]arene macrocycle **12** during redox stimuli and the structure of testing guest G**12**[42]. **c** the chemical structure of hydroxyphenyl-appended macrocycle **13** and the diversified chromism of rcct-**13** during stepwise oxidation and acidification. [O] denotes oxidation; [H+] and [B−] denote acid and base, respectively[43].

*Redox-RMs.* Redox is another stimulus that should be considered to design RMs because the charge and electron distribution of the macrocycle could be reversibly switched between the oxidized and reduced state. This allows us to dynamically control the chemical properties of the macrocycle.

Stoddart's group has extensively investigated a π-electron-deficient tetracationic cyclophane, cyclobis(paraquat-phenylene) (CBPQT[4+]) during the past two decades[39]. The CBPQT[4+] is an electron-deficient macrocycle that is a perfect recognition site, binding some π-electron-rich molecules, such as well-known tetrathiafulvalene (TTF) and 1,5-dioxynaphthalene (DNP)[40]. The CBPQT[4+] is also a redox-responsive macrocycle that could be reduced into diradical and dicationic form CBPQT[2(•+)]. The CBPQT[2(•+)] could form stable inclusion complexes with methyl viologen radical cations (MV[•+]) through radical-pairing interactions. The CBPQT[2(•+)] could be further reduced into a neutral form CBPQT. This time, the macrocycle CBPQT became

electron-rich and was able to bind π-electron-deficient guests, such as 1,4-dicyanobenzene (DCB) and 1,4-dicyanotetrafluorobenzene (DCFB) (Fig. 3a)[41]. These results clearly suggest that the electronic property of the macrocycle could be precisely controlled in multiple states. The binding ability to guest molecules with different electronic properties could also be achieved, being of great significance for the construction of cascade supramolecular systems.

Benzoquinone is a redox-responsive unit that can be reduced to electron-rich hydroquinone. Reversibly, the initial electron-poor benzoquinone can be recovered by oxidation. In 2016, Ogoshi et al. reported a redox-responsive pillar[5]arene macrocycle **12**, which contains one benzoquinone unit (Fig. 3b)[42]. The π-electron density of benzoquinone-containing macrocycle **12** was significantly lower than the classical pillar[5]arene. When the benzoquinone-contained macrocycle **12** was reduced into a hydroquinone-containing macrocycle **12**-2H, the inner cavity of

the macrocycle became electron-rich, showing a strong association with a redox-inactive guest molecule G12. The reversible host–guest complexation could be switched many times through the sequential addition of oxidant and reductant.

Instead of designing a RM by implanting active units into its backbone, the Hill's group appended the active units on the frame of a persistent macrocycle. Guided by the principle of redox switching between benzoquinone and hydroquinone, they designed a macrocycle 13 with the choice of 3,5-di-*t*-butyl-4-hydroxyphenyl (DtBHP) as the appendix and resorcinarene as the scaffold[43]. The resorcinarenes usually have four isomeric structures (rctt, rccc, rcct and rtct), depending on the relative orientations of substituents at their *meso*-positions. The authors prepared a pure isomer rctt-13. It is worth noting that the phenol groups of DtBHP could be facilely oxidized into hemiquinone groups by 2,3-dichloro-5,6-dicyano-1,4-benzoquinone (DDQ) or photochemical oxidation (UV 285 nm). The initial isomer rctt-13 was firstly oxidized to rctt-13-[Ox$_1$] by DDQ. Its configuration changed to rccc-13-[Ox$_2$] during the further step of oxidation (Fig. 3c). It is known that the reductive DtBHP phenol and oxidative DtBHP hemiquinone could form a 'hemiquinhydrone' complex by charge-transfer (C-T) interactions under acidic conditions. Thus, oxidative rctt-13-[Ox$_1$] and rccc-13-[Ox$_2$]

could be transformed to CT-rctt-13-[Ox$_1$] and CT-rccc-13-[Ox$_2$] by forming intramolecular 'hemiquinhydrone' pairs under acid stimulus. Thus, the initial rctt-13 macrocycle had five distinguishable states under continuous redox- and pH-stimuli. This multi-state macrocycle would be promising in applications for molecular logic gate operations, macular information storage, and smart chemosensors.

**Multistimulus RMs**. Up to now, most RMs have been responsive to a single stimulus. However, biological activities are often feasible under multiple stimuli, such as temperature, pH, light and small molecules. Thus, it is essential to design macrocycles whose responsiveness could be initiated by using orthogonal stimuli. Unfortunately, a limited number of examples of multistimulus RMs have been reported.

Otto et al. developed an "ingredients" approach to the design of a multistimulus RM[44]. By combining elements such as Azo, carboxylate, and thiol groups into the design of a building block, the authors identified a dimeric macrocycle 14 linked by disulfide bonds from dynamic combinatorial libraries. The macrocycle 14 coordinated with Mg$^{2+}$ into a hydrogel (Fig. 4a) that could be facilely switched between gel and solution by light, redox

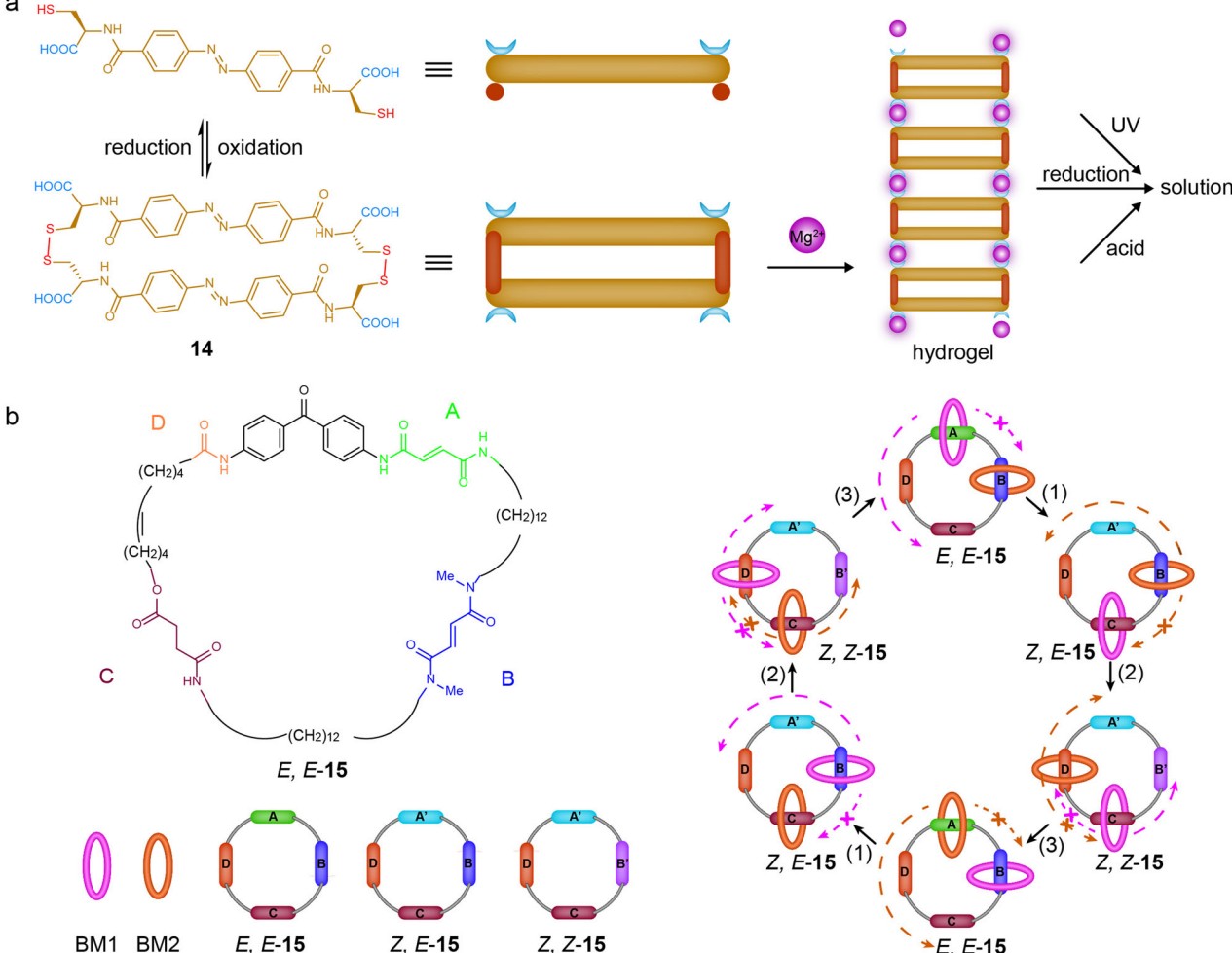

**Fig. 4 Multistimulus-responsive macrocycles. a** the multistimulus-responsive sol-gel process of the hydrogel formed by the macrocycle **14** and Mg$^{2+}$, [44]. **b** the chemical structure of macrocycle **15** containing four distinguishable recognition sites and the unidirectional rotation of two benzylic macrocycles (BM1 and BM2) along the track of the **15** in the [3]catenane controlled by alternate light irradiation and heating cycles: (1) 350 nm, 5 min, (2) 254 nm, 20 min and (3) 100 °C, 24 h[45].

reactions, pH, temperature, mechanical energy and sequestration or addition of Mg (II) salt. These results suggested that the dynamic combinatorial chemistry based "ingredients" approach provided an effective way to design and explore materials responsive to multiple stimuli.

Apart from the above "ingredients" approach of integrating different types of stimuli-responsive sites into a single macrocycle, the macrocycle itself could also be multistimulus responsive if the same input unit is capable of responding at specific modes (i.e. orthogonal light or temperature value). In 2003, Leigh and Wong et al. reported a [3]catenane-based molecular motor assembled by two identical benzylic amide macrocycles (BM1 and BM2) and a bigger macrocycle **15** with four distinguishable recognition sites (A: a secondary amide fumaramide group; B: tertiary amide fumaramide group; C: succinic amide ester group; D: isolated amide group) (Fig. 4b)[45]. The binding priority order of the benzylic amide macrocycles to the four recognition sites of the **15** was A > B > C > D. Thus, the two cyclic amide macrocycles (BM1 and BM2) were located at the A and B stations, respectively. Subsequently, when the stronger binding site *trans* form of the A station was transformed into the weaker binding site *cis* A' station under UV (350 nm) irradiation, macrocycle BM1 moved to the C station directionally, while the other pathway was blocked by macrocycle BM2. Similarly, followed by irradiation with 254 nm light, the stronger binding site *trans* form of the B station was transformed into the weaker binding site *cis* B', which caused macrocycle BM2 to shuttle to the D station directionally, while the opposite pathway was blocked by macrocycle BM1. A further step was triggered by heating at 100 °C. Both the *cis* A' and the *cis* B' were isomerized into initial *trans* A and *trans* B state, which rendered the macrocycle BM1, and BM2 moved back to the B and A stations. After undergoing another light irradiation and heating cycle, macrocycle BM1 and BM2 could return to the initial A and B stations through a directionally motor-like rotation. A similar system was reported in the same group in 2017[46]. An artificial [2]catenane-based rotary molecular motor containing a pH-RM was programmatically operated under the changeable pH values. Thus, the multistimulus responsiveness of the macrocycles could be programmed and precisely controlled, providing possibilities to implement more intricate tasks, such as biomolecular machines.

## Synthesis

The first step of making the design practical is to efficiently synthesize these delicate cyclic topologies. However, the synthesis of macrocycles is not easy due to the low yield in the final ring-closure step. This paper will review and discuss the ring-closure methods for the synthesis of RMs ranging from covalent coupling reaction and dynamic covalent chemistry to noncovalent self-assembly.

**Covalent coupling reactions for macrocyclization**. Covalent coupling reactions are the most popular methods for synthesizing RMs. Although the reactions always have low yields because the ring-closure process needs to overcome the energy of ring strain[47], the tough covalent bond stabilizes the macrocycles (Fig. 5a).

Nielsen's group applied the esterification reaction for the synthesis of a photo-RM by coupling an azo-derived acyl chloride with a dihydroazulene (DHA)-contained diol[48]. Linear polymers may be side products instead of the target macrocycle. To avoid polymerization, the reaction was performed at a low concentration with a slow feeding rate of reactants. The target product consisting of two isomers was obtained in yield of about 30%. Another dehydration reaction amidation was utilized to synthesize the aforementioned macrocycle **2** by using reactants of an azo-based acyl chloride and another azo-based amino[24].

Another significant covalent ring-closure protocol is derived from Stoddart's group. They have successfully explored a series of extended bipyridinium-based cyclophanes **16** for which the final ring close reaction took place between extended bipyridinium derivatives (ExⁿBIPY) and the open-chain dibromo-precursors (ExⁿDB²⁺)[39,49] (Fig. 5d). A similar cyclization reaction between diimidazole-modified pyridine precursors and dibromo-modified precursors was performed by Sessler's group to synthesize "Texas-sized" Box[26]. Prof. Xie's group synthesized a carbohydrate-based macrocyclic Azo by etherification between 2,2'-dihydroxyazobenzene and the dibromo-substituted sugar precursor[50]. The ring-closure reaction was implemented in the presence of 18-crown-6 catalyst in the dark, which gave rise to the best yield of 35%. Suzuki coupling reaction was also applied to connect bromo-substituted hemithioindigo units and borate-substituted biaryl sites in a yield of 30%[51]. We only summarized the dehydration reaction, pyridine or imidazole-based electrophilic substitution reaction, etherification and Suzuki coupling reaction here for the synthesis of RMs. However, much more ring-closure reactions have been reviewed for the general synthesis of macrocycles[8,52]. These reactions should be considered to synthesize new RMs in the future.

**Dynamic covalent chemistry for synthesizing RMs**. Dynamic covalent chemistry refers to chemical reactions that occur in a reversible manner[53]. The formation and break of the dynamic covalent bond could reach an equilibrium at the reaction condition. The product distribution highly depends on the thermodynamic stability of the resulting species. During the past decades, dynamic covalent reaction has been employed to link building blocks together for the synthesis of macrocycles. Moreover, as the reversible conversion of the dynamic reaction could be controlled by changing experimental conditions, the macrocycles connected by dynamic covalent bonds are responsive and switchable as well. Disulfide and hydrazone exchanges are the two types of reaction most used to construct RMs (Fig. 5b).

Disulfide bonds are very common in biological systems. The formation of these bonds is helpful for the folding and stability of proteins[54]. It could be produced by the oxidation of thiols in aqueous media at weak basic pH. Dynamic disulfide exchange occurs in the presence of a catalytic amount of thiol. Once all the thiols in the reaction are fully oxidized, the dynamic exchange will be stopped. When two thiol groups are decorated on a building block, a disulfide-linked macrocycle will form. The oxidation of dithiol building blocks is very efficient, and the synthesis of the disulfide macrocycles is in quantitative yields when the concentration of the building blocks is in the millimolar range[44]. Although the size distribution of the disulfide macrocycles is typically not selective in nontemplated libraries, disulfide chemistry could provide a solution to the ring-closure synthesis challenges of macrocyclic compounds and make the obtained macrocycles redox-responsive. By introducing a template molecule (N-methylated morphine) into the dithiol reaction mixture, a specific size of macrocycle **17** could be amplified quantitively (Fig. 5e), this was first noticed by Otto and Sanders, and it triggered a profound development of disulfide-based dynamic combinatorial chemistry[55]. For readers interested in dynamic combinatorial chemistry, please refer to the recent review[56].

Hydrazone chemistry is another type of dynamic covalent bond that could be used for the linkage of RMs. However, its reversible manner is different from disulfide chemistry. The hydrazone bond is formed by a condensation reaction between an aldehyde and a hydrazine with a molecular loss of water. If the

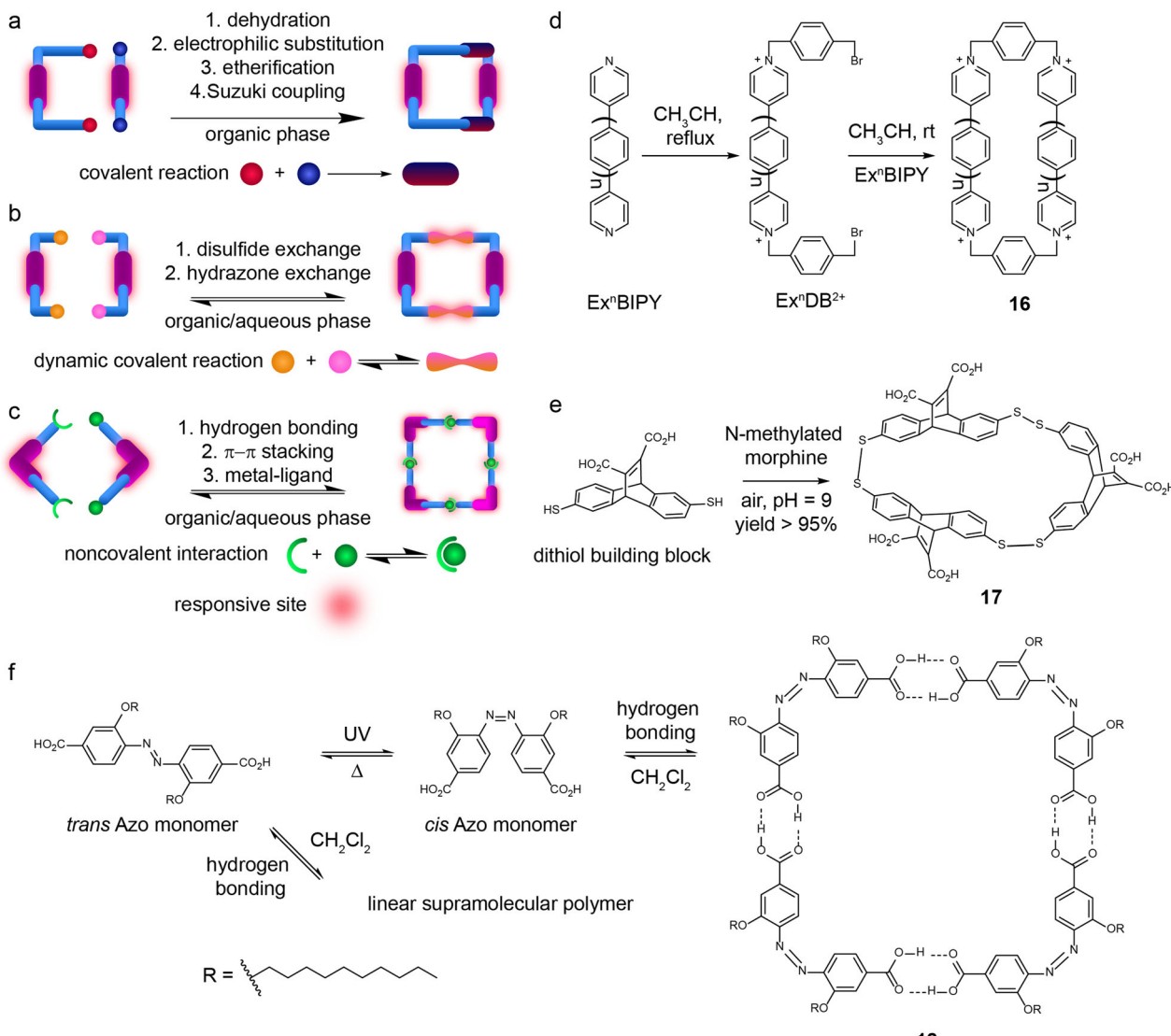

**Fig. 5 The schematic representation of ring-closure methods for RMs.** the RMs are synthesized by **a** covalent ring-closure strategy mainly including dehydration, electrophilic substitution, etherification and Suzuki coupling reaction in organic phase, **b** dynamic covalent ring-closure strategy mainly including disulfide and hydrazine exchange and **c** noncovalent ring-closure strategy mainly including hydrogen binding, π-π stacking and metal-ligand interactions. The synthetic route of **d** macrocycle **16** using the covalent ring-closure strategy[39,49], **e** macrocycle **17** using the dynamic covalent ring-closure strategy[55] and **f** macrocycle **18** using the noncovalent ring-closure strategy[61].

reaction is performed in water, the reaction becomes reversible, as the hydrazone bond can be hydrolysed. Thus, the formation of hydrazone bonds could not be a yield of 100% in water, which is less efficient than the formation of disulfide bonds.

The hydrazone chemistry is pH sensitive. Its optimal exchange kinetics at around pH 4.5 in water, which is not biocompatible. To extend its application in biological systems, aniline is often used as a catalyst for the reaction at the physiological pH[57]. Moreover, acyl hydrazones are photo-switchable. Photo-irradiation could drive $E \rightarrow Z$ isomerization in acyl hydrazones[58]. The reverse reaction can proceed either thermally or photochemically at a different wavelength. The pH- and photo-responsiveness made the hydrazone chemistry attractive in preparing RMs[38,59,60].

**Noncovalent self-assembly for making RMs.** By noncovalent interactions, such as hydrogen bonding, π-π stacking and metal-ligand coordination, building blocks with well-designed

topologies could be held together forming ring-like supramolecular structures (Fig. 5c). As the noncovalent forces are usually weak, the resulting self-assemblies are labile and sensitive to stimuli like pH, temperature and specific chemicals. In this section, emphasis will be given to the design of monomers that take part in molecular self-assembly.

Hydrogen bonding is a noncovalent linkage that has been utilized to design RMs. In 2003, Sleiman et al. reported a photo-RM **18** whose formation was directed by hydrogen bonding[61]. The authors first synthesized an Azo monomer equipped with two carboxylic acid groups. The *trans* form of the Azo could self-assemble into a linear supramolecular polymer through hydrogen bonding between the carboxylic acid groups. Under UV irradiation, the *trans* form was switched into the *cis* form, which further self-assembled into cyclic tetramer **18** (Fig. 5f).

The π–π stacking interaction between aromatic segments is another noncovalent force that took significant roles in making RMs. Lee and colleagues reported a noncovalent hexameric macrocycle consisting of six hydrophilic 120° V-shaped aromatic

monomers through π–π stacking interactions[62]. The macrocycle could further self-assemble into supramolecular nanotubes via mutual π–π stacking and van der Waals interactions. Interestingly, the nanotubes could undergo a reversible contraction-expansion through molecular slide among adjacent aromatic segments upon heating. These results suggest that the RMs by noncovalent bonding guarantees not only their dynamic nature but also considerable stability under external stimuli.

Discrete macrocycles could be constructed by coordination-driven self-assembly as well. Stang and co-workers have widely developed metallacycles with fantastic topology and function[31,63]. They reported a metallacycle that was almost quantitatively synthesized through the coordination between the *cis* stilbene-functionalized dipyridyl donor and an equivalent 180° di-Pt(II) acceptor[64]. When the initial *cis* stilbene units were isomerized into *trans* stilbene units after reversible photoisomerization, the discrete metallacycles transformed into linear metallosupramolecular polymers. This work represented a facile strategy to construct RMs with high efficiencies and showed the fragility of noncovalent-based metallacycle topology under photoisomerization.

## Application

As discussed, macrocycles have been proven to be an especially useful type of building block in various fields due to their unique structural properties. However, compared with the conventional shape-persistent macrocycles, RMs are more versatile to tune the physical and chemical properties of the cavity by using appropriate external stimuli. Thus, the dynamic nature of RMs would top up more functions and applications. Here the latest examples of application of RMs will be highlighted in the direction of guest encapsulation and controlled release (such as drug delivery, bioimaging and capture for pollutants or explosives), smart catalyst and controlling the properties of bioactive substance.

**Guest encapsulation and controlled release**. On one hand, the cavity of RMs could be employed to encapsulate guest molecules[24–26,36,38,65], which helps them work as nanocarriers. On the other hand, the size and property of the cavity could be adjusted to release the captured guests.

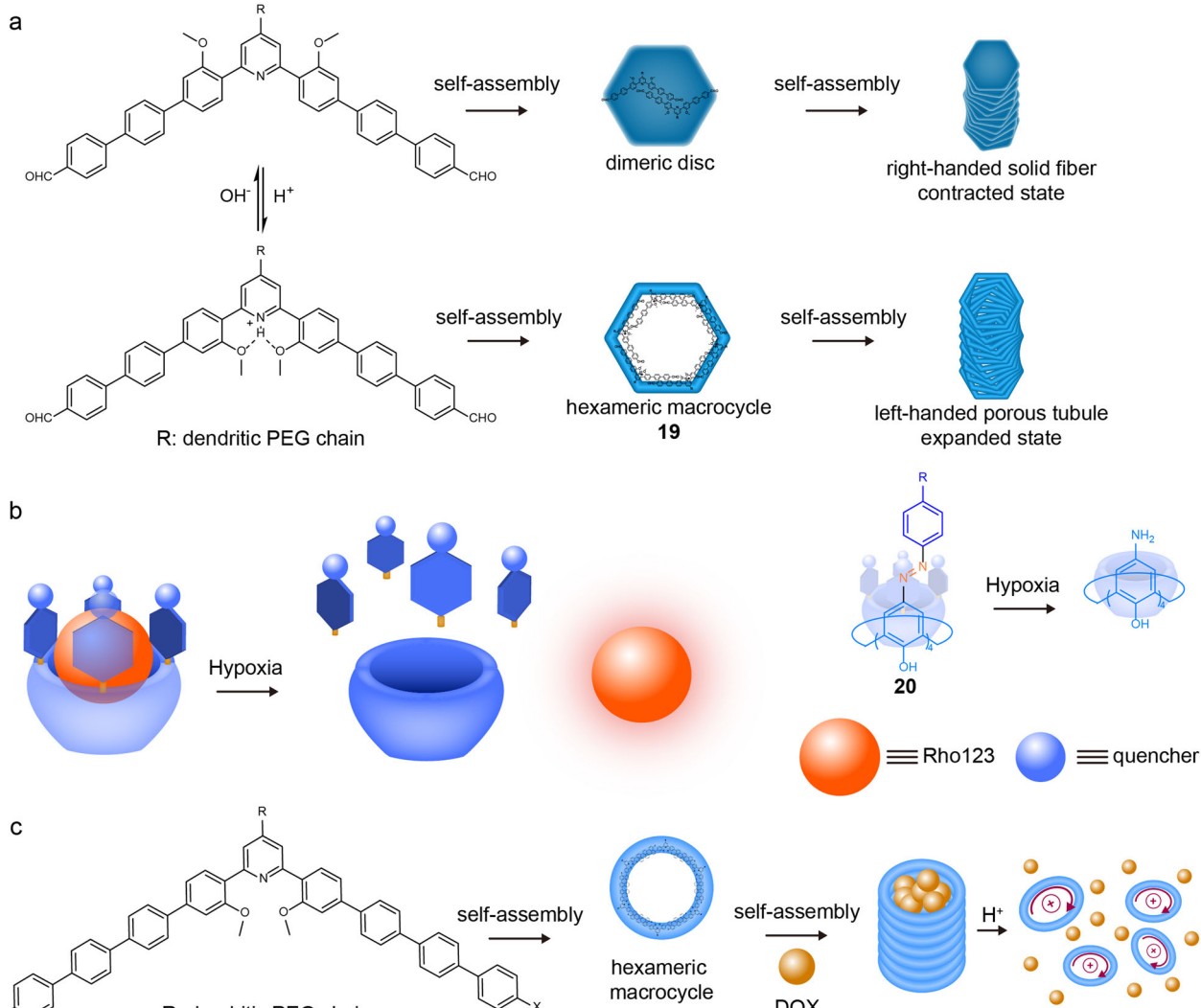

**Fig. 6 RMs for guest encapsulation and controlled release. a** the chemical structure of bent-shaped aromatic ligand, the self-assembly process under reversible acid/base-induced isomerization to form hexameric macrocycle **19** from dimeric disk and the further morphology changing process from right-handed solid fibers to left-handed hollow tubules[67]. **b** the chemical structure of the macrocycle **20** and the representation of hypoxia-responsive imaging mechanism[68]. **c** the chemical structure of the macrocycle **21**, the disassembly and further drug release process of the formed nanotubes after protonation of pyridyl site during the environmental pH changes[70].

Due to their specifically hollow nature, macrocycles served as ideal adsorbents for capturing harmful molecules[66]. Though shape-persistent macrocycles have exhibited a high capacity for pollutants, desorption remains a big challenge due to their mutual stable interaction. Huang et al. reported a macrocycle-based nanopump system[67]. The nanopumps were composed of bent-shaped aromatic ligands by π–π stacking interactions, which initially presented right-handed solid fibres by the stacking of dimeric ligands. After a protonation of ligands, the solid fibres were transformed into left-handed hollow tubules consisting of inflated hexameric macrocycles 19 (Fig. 6a). The porous tubules could efficiently capture organic pollutants, such as ethinyloestradiol (EO), bisphenol A (BPA) and methyl orange (MO) from wastewater. The subsequent desorption was precisely implemented through acid-triggered tubular contraction.

The dynamic process of encapsulation-release could also find its place in stimuli-responsive fluorescence probes. Guo et al. reported a redox-RM probe 20 for hypoxia imaging developed from a calix[4]arene appended with four 4-(phenylazo)benzoic acid groups (Fig. 6b)[68]. The fluorescence indicator Rhodamine 123 could fit well in the cavity of the macrocycle 20, which was suggested by a distinct fluorescent quench. As the Azo bond could be reduced to aniline derivatives under a reductive microenvironment, the macrocycle 20 was reductant sensitive. Interestingly, the Azo bonds of macrocycle 20 could be reduced in the hypoxia environment of cancer cells, which allowed the release of Rhodamine 123, thereby turning on its fluorescence for hypoxia imaging.

The objective of the drug delivery field is to develop smart nanocarriers that can load and release the drug molecules efficiently and precisely. RMs should be an ideal candidate for this because it has been proven that the captured guest by RMs could be released upon stimulation at a desired time. According to previous work[62,69], Huang and Shen et al. designed the π–π stacking macrocycles 21 for drug delivery[70]. The 21 could self-assemble into nanotubes through noncovalent interactions (Fig. 6c). The authors further discovered that the nanomaterials were biocompatible in vivo and could be used to load an anticancer drug doxorubicin (Dox). Dox-loaded nanocarriers 21 exhibited higher antitumour efficiency and lower systemic toxicity in vivo compared to free Dox. The release mechanism was explained as: the pyridine-containing macrocycles 21 were selectively protonated by the acidic environment of the tumour site, which allowed the dissociation of nanotubes by electrostatic repulsion, thereby releasing the included Dox.

**Smart catalysis**. Massive chemical reactions in living systems are almost catalysed by enzymes. However, catalytic performance has not been fully understood, as the catalysis is always accompanied by the continuous regulation of the enzyme. Inspired by nature, chemists have started to mimic such dynamic processes and develop smart catalytic systems in which the reaction progress can be precisely controlled under appropriate stimuli[71]. Particularly, RMs have been demonstrated as efficient smart catalysts.

In 2014, Otto's group designed a smart catalytic system from dynamic combinatorial libraries (DCLs)[72]. DCLs prepared from a dithiol building block in water were oxidized and initially dominated by interlocked [2]catenanes (Fig. 7a). The dominated [2]catenanes could be disassembled into a tetrameric macrocycle 22 in the presence of a cation template. The template was employed as the substrate of the intramolecular aza-Cope rearrangement that was selected as a model reaction. During

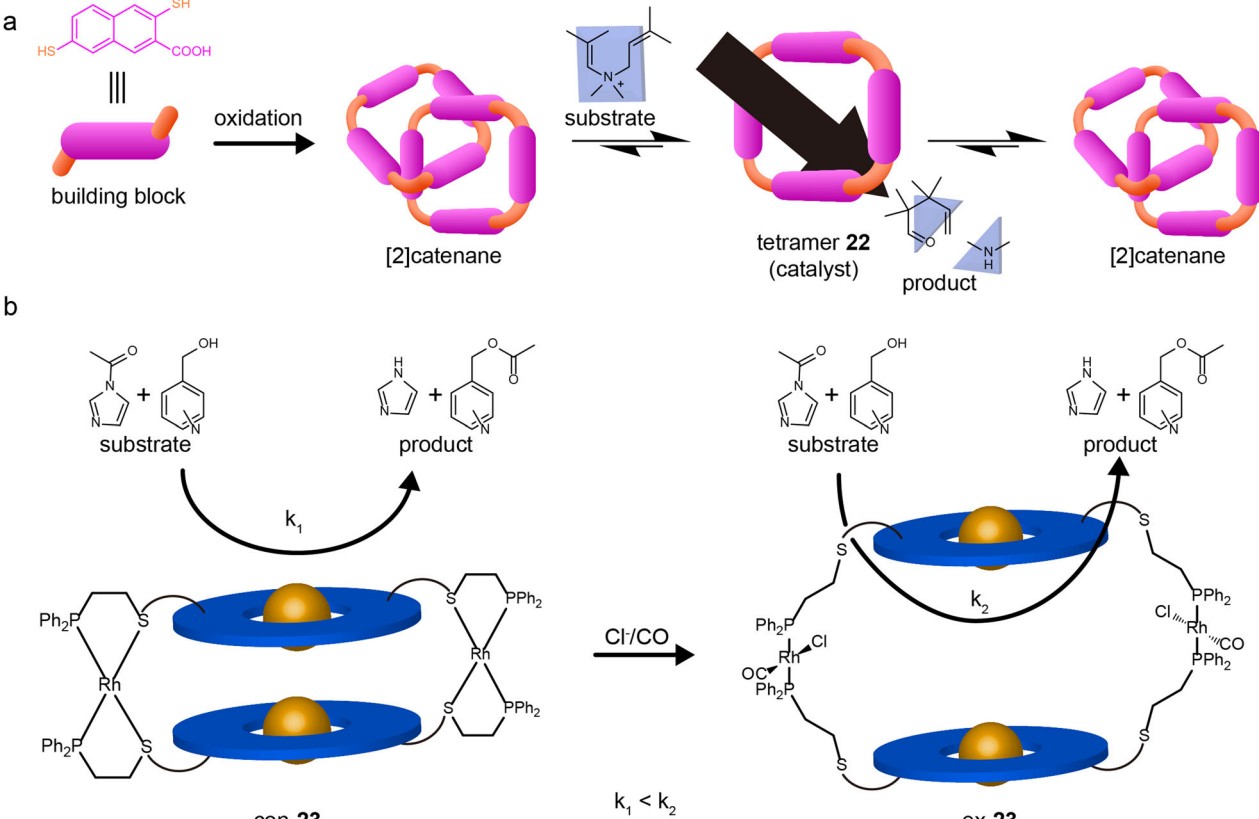

**Fig. 7 SRMs for smart catalyst. a** oxidation of dithiol building block and the dynamic catalyzed reaction between cation substrate and catalyst tetrameric macrocycle **22**[72]. **b** the catalytic process of acyl transfer reactions by a closed macrocycle con-**23** vs open macrocycle ex-**23**, the latter can bind the substrates within the bigger cavity renders the higher rate of the reaction[73].

the oxidation process, the substrate was gradually consumed to produce the rearrangement product. When the substrates were completely converted, catalyst **22** was also consumed to produce the more thermodynamic stable species [2]catenanes in the disulfide molecular network.

Also inspired by the enzymatic regulation in nature, Mirkin and Nguyen et al. reported a RM-based catalyst **23** that was coordinated by porphyrin-based thioether-phosphine hemilabile ligands and transition-metal precursors ($Rh^I$) (Fig. 7b)[73]. The condensed state of the macrocycle con-**23** could be switched into an expanded macrocycle ex-**23** by introducing benzyltriethyl ammonium chloride (BTAC) and CO. Herein, a catalytic acyl transfer reaction was employed as the research model. As the substrates could be inclusive in the cavity of the con-**23**, they could be converted to the products in a catalytic fashion. Interestingly, when the con-**23** was transformed into the ex-**23**, the reaction rate was greatly accelerated ($k_2 > k_1$), suggesting that the catalytic activity could be effectively regulated by changing the size of the macrocyclic catalyst.

**Controlling the properties of bioactive substances.** Cellular processes are extraordinarily complex, requiring strict spatiotemporal organization by well-tuned signaling networks, in which peptides are efficient biological modulators and promising drugs. Through artificial synthetic methods, stimulus-responsive units can be introduced into bioactive molecules to give rise to controllable functionalities. Specifically, cyclic compounds play significant roles in living organisms due to their unique topologies, which have been proven to be promising drug candidates[74].

Cyclic peptides and peptidomimetics have attracted much attention in the field of drug discovery[75]. Ulrich and co-workers synthesized a photo-controllable cyclic peptidomimetic **24** for cancer therapy[76]. The **24** was designed by the incorporation of a photo-switchable diarylethene (DAE) unit into the cyclic structure of the natural peptide antibiotic gramicidin S (GS) (Fig. 8a). The peptide GS is known to be highly cytotoxic to not only tumour cells but also normal cells due to its unique structure. Significantly, before the photoisomerization of the DAE unit, the closed-form **24** was 5.5- to 8.0-fold lower in cytotoxicity for normal cell than open-form **24**, which possessed toxicity against cancer cell similar to peptide GS. Thus, the cytotoxicity of the **24** could be reversibly controlled during cancer therapy.

Cell penetration is another significant factor affecting the efficacy of peptide-based drugs. It could be regulated by using RMs. Lee and Yu et al. reported an Azo-based amphipathic

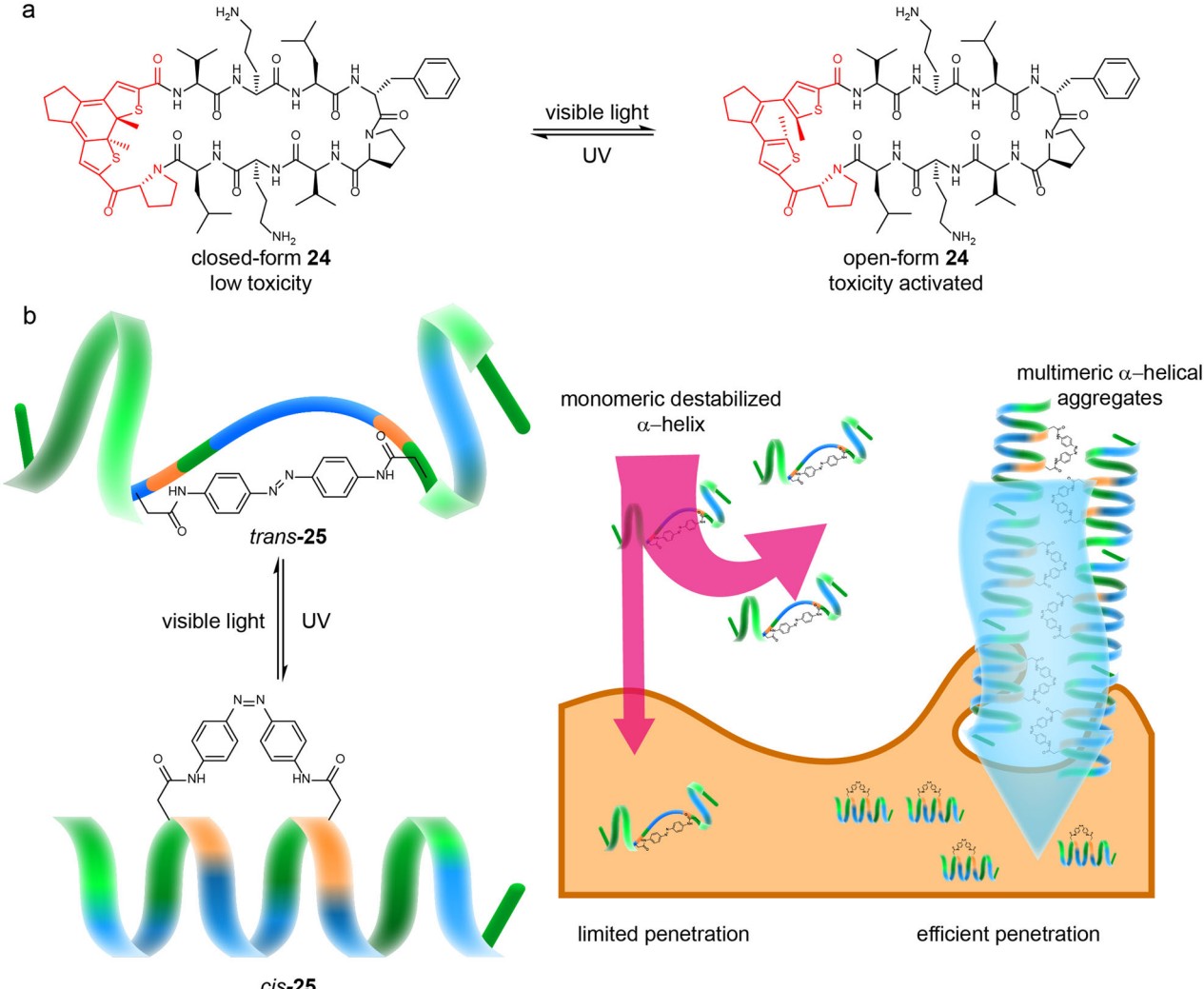

**Fig. 8 SRMs for controlling the properties of bioactive substance. a** the chemical structure and switchable cytotoxicity of DAE-based peptidomimetic macrocycle **24** during the reversible photoisomerization[76]. **b** photoswitched helical structure of azo-based cyclic peptide **25** and the further self-assemble into multimeric α-helical aggregates for enhancing cell penetration[78].

peptide whose-cell penetration could be regulated by *cis–trans* isomerization of the Azo unit[77]. Previous studies revealed that most peptide-based drugs possess poor permeability and instability in vivo[78]. Here, a photo-responsive Azo unit was incorporated into an amphipathic peptide to form a cyclic conformation **25**. An α-helical conformation of peptide **25** could be formed after the *trans* isomer was turned to the *cis* isomer. Importantly, the α-helical peptides *cis*-**25** could self-assemble into multimeric α-helical aggregates (Fig. 8b). Further cell experiments showed that cell penetration and stability were obviously enhanced compared to monomeric *trans* isomers.

## Conclusions and outlook

In summary, we have given a review of the emerging field of RMs. First, the main design strategy for RMs was discussed. In most cases, functional units that are responsive to external stimuli, such as light, pH and redox, can be incorporated into the backbone of macrocycles. RMs can also be designed by appending the responsive units to the macrocycles. Up to now, most RMs have been responsive to a single stimulus. There are few examples of RMs that performed dynamics under multiple stimuli. In the future, it is essential to design macrocycles whose responsiveness could be initiated by using orthogonal stimuli to spatio-temporally control the property of the macrocycle. For example, we could develop macrocycles that are photo-responsive at different wavelengths. Currently, the number of responsive types is quite small. Temperature-, magnetic- and mechano-sensitive moieties can also be used to enrich the toolbox for design.

In addition, we have summarized the latest attempts that deal with the long-standing challenges of the ring-closure step in the synthesis of RMs. Cyclization by using covalent synthesis is a traditional approach to prepare a macrocycle, but it always has low yields and reqires time-consuming purification. Recently, the dynamic covalent chemistry has emerged as a powerful strategy to make RMs because such chemical bonds are reversible and responsive. However, this advantage may bring a short-coming (i.e. low yields) to the reaction. This drawback could be overcome by using the tool of dynamic combinatorial chemistry. The template-directed synthesis from dynamic combinatorial libraries could help to improve the yield. There have already been examples showing quantitative synthesis, which will ease the purification step. Molecules can also noncovalently self-assemble into ring-like structures. RMs of this type are always labile, and it may also be challenging to characterize the exact number of monomers in the ring. The structure of channel-like proteins could provide inspiration to change the situation of deficiency. If relatively larger monomers can be designed that contain many units for noncovalent interactions, stable yet responsive structures of self-assembled macrocycles may be obtained.

Finally, the latest application of RMs has been highlighted. RMs have been used to bind and release guest molecules for drug delivery, bioimaging and removing pollutants. They have also been applied for smart catalysis and controlling the properties of bioactive substances. However, there are still major challenges that need to be overcome for further practical applications. For example, as binding and controlled release are key for application, it is of great importance to develop RMs with high specificity and affinity for tasks to be performed more accurately and efficiently, such as guest capture, detection and transportation. Furthermore, the application always needs research backgrounds from various fields. Multidisciplinary collaboration will more than likely be a powerful working horse to produce functions and bring a bright future for science.

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

## Acknowledgements
We are grateful for the financial support from the National Natural Science Foundation of China (21801052), Hainan University start-up fund (KYQD(ZR)1852), the construction program of research platform in Hainan University (ZY2019HN09), the Finnish Culture Foundation, the Sigrid Jusélius Foundation and the Academy of Finland (Decision No.318524).

## Author contributions
J.L. conceptualized the paper and J.Y. searched the literature and D.Q. drew figures, J.L. and J.Y. wrote, proofread and edited the paper.

## Competing interests

The authors declare no competing interests.
