## [Peer Review File · Communications Chemistry]

REVIEWERS' COMMENTS:

Reviewer #1 (Remarks to the Author):

This review by Jianwei Li and co-workers provide an overview on the design, cyclization, applications and future of responsive macrocycles (RM). It can guide readers to learn about RM thoroughly. Thus, I would like to recommend its publication in Communications Chemistry after minor revisions suggested below.

1. Some relevant references can be included in reference 19 or reference 20, i.e., J. Am. Chem. Soc., 2019, 141, 1280-1289; J. Am. Chem. Soc., 2017, 139, 7168-717; Chem. Commun, 2019, 55, 13462-13465; ACS Appl. Mater. Interfaces. 2019, 11, 16117-16122; Adv. Sci., 2020, 7, 2000803; Chem. Commun. 2019, 55, 4499-4502; Angew. Chem. Int. Ed. 2017, 56, 7062-7065; Angew. Chem. Int. Ed., 2015, 54, 9376-9380.
2. In line 19 of page 2, the word "photo" should be replaced with "light".
3. In line 3 of page 4, the authors need to check the stoichiometric ratio to be 1:2 or 2:1.
4. The end page of reference 1 is missed, and reference 54 should be updated.
5. The chemical structures in Figure 1a,b,c are not clear.

Reviewer #2 (Remarks to the Author):

I was happy to read this nice piece of review article that summarizes recent advances of stimuli-responsive toroidal nanostructures. Considering the importance of toroidal geometry as a key component for the development of intelligent nano materials, I believe that this manuscript has a merit to be published in Commun. Chem. as a review article after considering the following minor points.

- 1) The chemical structures of 16-18 are missing. Please add these in the figures.
- 2) In Synthesis part, if the authors provide typical examples of macrocyclization as a figure, the readability of the manuscript will be highly improved.
- 3) The following references related to stimuli-responsive toroids are missing. (Acc. Chem. Res. 2013, 46, 2888-2897; Nat. Commun. 2019, 10, 1080)

Reviewer #3 (Remarks to the Author):

In this review article Li and coworkers summarized the design, synthesis and application of responsive macrocycles (RMs) in which stimuli-responsive moieties are incorporated within the macrocycle. Three types of RMs that are controlled by external stimuli including light, pH and redox were highlighted. This review provided useful information on adaptive macrocyclic hosts, therefore it could be published after addressing the following comments.

1. "confirmation selection model" should be "conformation selection model".
2. change "such as photo, pH, and redox" to "such as light, pH and redox".
3. The structures of 16-18 are missed in the text.
4. Figure 5c, the supramolecular interaction is too general, I understand it should be noncovalent bond.
5. Figure 5c, what the "responsiveness" mean?

.....

Point-to-Point Response to Reviewers' Comments

Reviewer #1 (Remarks to the Author):

This review by Jianwei Li and co-workers provide an overview on the design, cyclization, applications and future of responsive macrocycles (RM). It can guide readers to learn about RM thoroughly. Thus, I would like to recommend its publication in Communications Chemistry after minor revisions suggested below.

1. Some relevant references can be included in reference 19 or reference 20, i.e., J. Am. Chem. Soc., 2019, 141, 1280-1289; J. Am. Chem. Soc., 2017, 139, 7168-717; Chem. Commun, 2019, 55, 13462-13465; ACS Appl. Mater. Interfaces. 2019, 11, 16117-16122; Adv. Sci., 2020, 7, 2000803; Chem. Commun. 2019, 55, 4499-4502; Angew. Chem. Int. Ed. 2017, 56, 7062-7065; Angew. Chem. Int. Ed., 2015, 54, 9376-9380.

Answer: The related literatures have been added in the revised manuscript.

2. In line 19 of page 2, the word "photo" should be replaced with "light".

Answer: The modification has been made in the revised manuscript.

3. In line 3 of page 4, the authors need to check the stoichiometric ratio to be 1:2 or 2:1.

Answer: The related errors have been corrected in the revised manuscript.

4. The end page of reference 1 is missed, and reference 54 should be updated.

Answer: The related references have been added and updated in the revised manuscript.

5. The chemical structures in Figure 1a,b,c are not clear.

Answer: We have improved the related Figure resolution in the revised manuscript.

Reviewer #2 (Remarks to the Author):

I was happy to read this nice piece of review article that summarizes recent advances of stimuli-responsive toroidal nanostructures. Considering the importance of toroidal geometry as a key component for the development of intelligent nano materials, I believe that this manuscript has a merit to be published in Commun. Chem. as a review article after considering the following minor points.

1) The chemical structures of 16-18 are missing. Please add these in the figures.

Answer: The related chemical structures have been added in Figure 5 in the revised manuscript.

2) In Synthesis part, if the authors provide typical examples of macrocyclization as a figure, the readability of the manuscript will be highly improved.

Answer: The typical examples have been added in Figure 5 in the revised manuscript.

3) The following references related to stimuli-responsive toroids are missing. (Acc. Chem. Res. 2013, 46, 2888-2897; Nat. Commun. 2019, 10, 1080)

Answer: The related literatures have been added in the revised manuscript.

Reviewer #3 (Remarks to the Author):

In this review article Li and coworkers summarized the design, synthesis and application of responsive macrocycles (RMs) in which stimuli-responsive moieties are incorporated within the macrocycle. Three types of RMs that are controlled by external stimuli including light, pH and redox were highlighted. This review provided useful information on adaptive macrocyclic hosts, therefore it could be published after addressing the following comments.

1. "confirmation selection model" should be "conformation selection model".

Answer: The modification has been made in the revised manuscript.

2. change "such as photo, pH, and redox" to ""such as light, pH and redox".

Answer: The modification has been made in the revised manuscript.

3. The structures of 16-18 are missed in the text.

Answer: The related chemical structures have been added in Figure 5 in the revised manuscript.

4. Figure 5c, the supramolecular interaction is too general, I understand it should be noncovalent bond.

Answer: The related expressions have been changed into "noncovalent interaction" in the revised manuscript.

5. Figure 5c, what the "responsiveness" mean?

Answer: The "responsiveness" maybe not clear, we have changed this word into "responsive site", which represents the responsive groups or bonds (such as azobenzene group, disulfide bond or hydrogen bond).